# miR-221-5p and miR-186-5p Are the Critical Bladder Cancer Derived Exosomal miRNAs in Natural Killer Cell Dysfunction

**DOI:** 10.3390/ijms232315177

**Published:** 2022-12-02

**Authors:** Ting Huyan, Lina Gao, Na Gao, Chaochao Wang, Wuli Guo, Xiaojie Zhou, Qi Li

**Affiliations:** School of Life Sciences, Northwestern Polytechnical University, Xi’an 710072, China

**Keywords:** bladder cancer (BC), natural killer (NK) cells, exosome, miRNA, miR-221-5p, miR-186-5p

## Abstract

Bladder cancer (BC) is the tenth most commonly diagnosed cancer worldwide, and its carcinogenesis mechanism has not been fully elucidated. BC is able to induce natural killer (NK) cell dysfunction and escape immune surveillance. The present study found that exosomes derived from the urinary bladder cancer cell line (T24 cell) contribute in generating NK cell dysfunction by impairing viability, and inhibiting the cytotoxicity of the NK cell on target cells. Meanwhile, T24 cell-derived exosomes inhibited the expression of the important functional receptors NKG2D, NKp30, and CD226 on NK cells as well as the secretion of perforin and granzyme-B. The critical miRNAs with high expression in T24 cell-derived exosomes were identified using high-throughput sequencing. Furthermore, following dual-luciferase reporter assay and transfection experiments, miR-221-5p and miR-186-5p were confirmed as interfering with the stability of the mRNAs of DAP10, CD96, and the perforin gene in NK cells and may be potential targets used in the therapy for BC.

## 1. Introduction

Bladder cancer (BC) is one of the most prevalent cancers, with a high incidence and fatality rate [1]. Despite receiving years of attention and study, the mechanism of BC oncogenesis remained unexplained. Multiple structural and genetic abnormalities, as well as variations in cellular metabolism and intracellular signaling pathways, were postulated to contribute to the oncogenesis of BC [2]. On the other hand, insufficient anti-tumor immunosurveillance and the ability of malignancies to evade immune clearance may partially explain the cause of oncogenesis. By gradually accumulating a set of mutations, tumors downregulate the MHC Class I expression on their surface to avoid being killed by CD8^+^ T cells [3]. By secreting tumor-related factors and extracellular vesicles (EVs) into the tumor microenvironment, tumors may also trigger the dysfunction and exhaustion of a variety of immune cells, such as CD4^+^ T cells, CD8^+^ T cells, natural killer (NK) cells, and dendritic cells (DC), [4,5,6,7]. Exhausted anti-tumor immune function has been reported in numerous tumors, including breast cancer, lung cancer, colon cancer [8], kidney cancer, pancreatic cancer, ovarian cancer [9], glioma, and melanoma [10]. Improving the immunotherapeutic efficacy and prognosis of cancer patients requires that the mechanism of anti-tumor immune exhaustion be clarified.

Independent of prior activation, NK cells (CD56^+^ CD3^−^) are a subset of large granular lymphocytes that can directly kill altered and viral-infected cells, and promote adaptive immunity by secreting pro-inflammatory cytokines and chemokines [11]. NK cells serve a vital role in anti-tumor immune surveillance; they are able to identify and eliminate tumor cells that have evaded T cell lysis, while enhancing the anti-tumor activity of T cells by secreting cytokines [12]. Based on the delicate balance between activating and inhibiting signals, NK cells are either activated or maintained in a quiescent state. According to clinical investigations, the decreased number and impaired activity of NK cells were detected in patients with BC [13,14] suggesting that BC cells can evade anti-tumor immune surveillance by inducing NK cell dysfunction via an unknown mechanism.

Exosomes are nano lipid bilayer vesicles with a 30–120 nm [15] diameter that are secreted by virtually all live cells and have been discovered in various body fluids [16]. Exosomes, and exosomal cargos such as DNA, RNA, lipids, and proteins, have garnered a great deal of attention due to their essential role in intercellular communication and regulation of target cells activity. Exosomes, which mirror the characteristics of their parent cells [17], have been utilized as biomarkers in disease diagnosis, prenatal sex determination, and the vaccines manufacture [18]. Tumor cells secrete roughly 10 times more exosomes than healthy cells [19,20,21]. These tumor-derived exosomes (TEXs) play a crucial role in carcinogenesis, development, metastasis, angiogenesis, and drug resistance. In recent years, the immunological regulating function of TEXs has been steadily revealed [22]. TEXs impaired T cell activity in a tumor microenvironment by reducing the expression of CD3 and JAK3 in primary activated T cells and by mediating Fas/FasL (Fas ligand)-driven CD8^+^ T cells death [23]. Upon exposure to TEXs, NK cells exhibit a similar reduction of activity as T cells. TEXs reduce the cytotoxicity of NK cells in myelogenous leukaemia patients via increasing SMAD phosphorylation and reducing NKG2D receptor expression [23]. Another study showed that clear cell renal cell carcinoma (ccRCC) can release exosomes to promote NK cells dysfunction and evade innate immune surveillance by regulating the TGF-β/SMAD pathway [24]. Additional evidence has confirmed the suppressed cytotoxicity capacity of NK cells in cancer patients, as described in the review of Wioletta et al. [25].

In the tumor microenvironment, there is an undeniable interaction between TEXs and NK cells, but the influence of BC-derived exosomes on NK cells has not been adequately investigated. The objective of this study is to investigate the effect of BC-derived exosomes on NK cells. After treating human primary NK cells with exosomes derived from a BC cell line (T24 cells), the cytotoxic activity, the expression of essential functional receptors, and the cytotoxic cytokines secretion of NK cells were measured. Furthermore, the miRNA signature of T24 cell-derived exosomes was profiled by using high-throughput sequencing, and the essential miRNAs that play a crucial role in NK cell activity were screened and verified.

## 2. Results

### 2.1. Exosome Identified

After the multistep centrifugation, the collected vesicles from T24 and SV-HUC-1 culture supernatant were morphologically analyzed by TEM. It was shown that these vesicles presented typical sphere- and saucer-like structures with the diameter between 30-120 nm (Figure 1A), and a mean diameter of 93.8 ± 37.5 nm which was detected by Zetasizer Nano ZS (Figure 1B). The WB results indicated the expression of the bio-markers of CD63, CD81, and Hsp70 on these vesicles (Figure 1C). This result suggested that the collected vesicle confirms the characteristics of exosomes.

### 2.2. NK Cell Identified

After 3 weeks in vitro expansion, by CD56 and CD3 antibodies that were double stained, the flow cytometry data indicated that the proportion of NK cells (CD56^+^CD3^−^) in cultured cells is 88.80 ± 1.80% (data from 5 donor), with little NKT (CD56^+^CD3^+^) cells (6.70 ± 0.85%) present (Figure 2).

### 2.3. NK Cell Can Take Up Exosome Efficiently

The image captured by inverted fluorescence microscope revealed that the majority of NK cells showed green signal which derived from labeled exosomes (Figure 3A–C), and the positive rate is about 91.6 ± 2.4% (Figure 3D).

### 2.4. The Effect of T24 Exosome on NK Cells

The T24 exosomes (T24-EXO) were used to treat NK cells for 24 h–48 h with the SV-HUC-1 exosomes (SV-EXO) treated group as control. The viability data showed that, compared to the SV-EXO group, T24-EXO impaired NK cell viability as well as inhibiting the cytotoxicity of NK cells on target cells (K562) after 48 h treatment. The OD_450_ value of NK cells in the T24-EXO group is 0.44 ± 0.09 compared with 0.61 ± 0.07 in control (Figure 4A). Meanwhile, the killing rates of 48 h treatment are 80.55% ± 3.43% (T24-EXO group) and 91.09% ± 1.94% (control) respectively (Figure 4B).

The apoptosis assay confirmed that the T24-EXO induced more NK cell apoptosis compared to the control after 48 h treatment. Early apoptosis and late apoptosis cells in T24-EXO treated group occupied 15.69% ± 2.62% and 4.13% ± 1.94% compared to 13.33% ± 3.19% and 4.23% ± 2.08% in the control group at 24 h, as well as 30.14% ± 8.94% and 9.83% ± 7.35% in T24-EXO group compared to 17.72% ± 9.02% and 5.06% ± 4.13% in the control group (48 h) (Figure 4C,D). Western blot results showed that downregulated perforin and granzym-B secretion of NK cells may be the reason for their lost cytotoxicity after T24-EXO treatment. There is no significant change of IFN-γ secretion between T24 and SV-HUC-1 exosomes treated NK cells (Figure 4E).

To further investigate the underlying mechanism by which T24-EXO inhibits NK cell cytotoxicity, the key functional receptors of NK cells were evaluated. The data of flow cytometry indicated that the NKG2D, NKp30, and CD226 are significantly affected by T24-EXO, the percentage of NKG2D, NKp30, and CD226 in T24-EXO group are 82.24 ± 5.61%, 56.21 ± 10.99%, and 81.47 ± 3.31%, respectively compared to 95.93 ± 6.18% (NKG2D), 88.56 ± 3.19% (NKp30), and 90.20 ± 2.07% (CD226) in control. Meanwhile, the MFI are 66.35 ± 6.04 (NKG2D), 21.96 ± 9.09 (NKp30) and 51.78 ± 6.65 (CD226) in T24-EXO group, compared to 84.19 ± 7.00 (NKG2D), 67.05 ± 8.65 (NKp30) and 70.25 ± 2.68 (CD226) in control. Nevertheless, the receptors of NKG2A, NKp44 and NKp46 showed no significant changes by T24-EXO treatment compared to the control in NK cells (Figure 5A–C).

### 2.5. The Effects of Exosomal miRNAs on NK Cells

Through calculating by the online programs, the miRNAs and their predicted target genes in NK cells were shown in Appendix A. Based on the results of calculation, miR-221-5p and miR-186-5p were chosen to be further analyzed. The results of dual-luciferase reporter assay showed that the gene of DAP10 (*HCST-001*) is the practical target gene of miR-185-5p, and the gene of CD96 (*CD96*) and perforin (*PRF1*) are the target genes of miR-221-5p (Figure 6).

Due to the low transfection efficiency of the routine method on NK cells, the SV-HUC-1 derived exosomes were employed to carry miRNA mimic and transfect NK cells. After 48 h transfection, the content of miRNA in NK cells was evaluated by the q-PCR method. It was shown that the content of miR-221-5p and miR-186-5p were significantly increased in NK cells, which indicated exosomes are an ideal carrier and can transport miRNA cargo to target cells efficiently (Figure 7A).

After 48 h transfection, the NK cells were harvested to examine the target genes in protein level. The result showed that, the DAP10, CD96 and perforin in NK cells were significantly downregulated. The Western blot data validated these results (Figure 7B).

## 3. Discussion

The mechanism of BC carcinogenesis is currently incompletely known. In addition to the accumulated variation in tumor cells, increasing experimental and clinical data suggest that tumor-induced immune dysfunction plays a crucial role in the occurrence and development of malignancies. NK cell dysfunction has been observed in numerous cancer patients, such as colorectal carcinoma [26], gastric cancer [27], pancreatic cancer [28], and chronic lymphocytic leukaemia [29]. NK cells play a critical role in BC as well. Hermann et al., found that there are decreased NK cell counts and impaired lymphokine-activated killer (LAK) cytotoxicity of PBMC from patients with noninvasive and invasive transitional-cell bladder cancer [30]. Similarly, Zhang et al., reported that, compared to the healthy volunteers, the circulating NK cells from bladder cancer patients exhibited significantly lower cytotoxicity. This is in part due to the over-representation of immunosuppressive Tim-3^+^ NK cells in bladder cancer patients [31]. The study from Neelam et al., showed that NK cells are the important intratumoral lymphocytes in BC, among them, CD56^dim^ NK cells were associated with higher pathologic stage, and CD56^bright^ NK cells are prognostically relevant to BC, making them a promising target for cancer therapy [32].

The accumulation of evidence suggests that in tumor microenvironment, tumor remodeling the function of immune cells via releasing TEXs [33]. For instance, tumor-derived exosomes contain soluble NKG2D ligands and TGF-β, which can inhibit the surface NKG2D expression and reduce NKG2D-dependent cytotoxicity on NK cells and CD8^+^ T cells [34]. In the present study, it was demonstrated that T24 cell-derived exosomes can also affect the activity of NK cells, as evidenced by the decreased vitality and impaired cytotoxicity of NK cells after T24 cell-derived exosomes treatment. The diminished NK cells cytotoxicity may be attributable to the downregulated activated receptors NKG2D, NKp30, and CD226, as well as the decreased production of perforin and granzyme B.

miRNAs regulate the expression of protein-coding genes in a sequence-specific way by inhibiting their translation or cleaving RNA transcripts. Recently, exosomal microRNAs have gained increasing interest due of their amazing biological effects. To date, aberrant expression of more than 200 miRNAs and miRNA families/clusters has been found in BC [35]. Using high-throughput sequencing, the important exosomal microRNAs with significant abnormal expression in T24 cells were discovered. Calculations revealed that miR-221-5p and miR-186-5p, are highly linked with the dysfunction of NK cells. The moderating impact of miR-186-5p on DAP10, and miR-221-5p on perforin and CD96 were validated by using dual-luciferase reporter and Western-blot assay.

miR-221-5p is typically dysregulated in cancer patients. The research of Liu et al. revealed that miR-221-5p is considerably elevated in clear cell renal cell carcinoma (ccRCC) tissues and ccRCC cell lines, and acted as a crucial oncogene [36] Similarly, it was discovered that miR-221-5p increases prostate cancer cell growth and metastasis [37]. However, other studies have demonstrated that miR-221-5p is downregulated in prostate cancer compared to normal prostate tissue [38], as well as in GC tissues and cell lines [39], suggesting that this microRNA may play negative roles in these tumors. miR-221-5p is significantly upregulated in T24 cell-derived exosomes compared to SV-HUC-1 exosomes, as demonstrated by our previous results [40]. It was claimed that each miRNA has a distinct expression pattern in different malignancies and various samples, such as tissue, body fluid, and extracellular vesicles. Other studies demonstrated the possible effect of miR-221-5p in immunologic processes, such as regulation of inflammatory responses in acute gouty arthritis by targeting IL-1 [41] and anti-inflammatory function in human colonic epithelial cells by targeting the interleukin 6 receptor (IL-6R) [42]. Our results demonstrated that miR-221-5p can target CD96 and perforin in NK cells, indicating that it is a promising target for intervention in BC immunotherapy.

miR-186-5p is also a well-studied miRNA that exhibits complex effects in tumors, such as playing a positive role in lung adenocarcinoma by targeting PTEN [43] and inhibiting tumor growth in osteosarcoma [44], non-small-cell lung cancer cells (NSCLCs) [45], colorectal cancer cell [46], and neuroblastoma [47]. In NK cells, miR-186-5p can target DAP10, an essential adapter protein of NKG2D, and inhibit NKG2D-DAP10 mediated activating signal in activating NK cell cytotoxicity. However, the cause of decreased expression of NKG2D, NKp30, CD226 and granzyme-B, as well as increased apoptosis in NK cells following treatment with T24 cell-derived exosomes remains to be further studied.

Exosomes were deemed excellent nucleic acid carriers due to their inherent qualities, and have been utilized by numerous researchers to deliver therapeutic nucleic acid medications to target cells [48]. For instance, Kamerkar et al. demonstrated that exosomes have a better capacity to transfer siRNA and shRNA to inhibit tumor growth in vivo [49]. In this study, we attempted to use SV-HUC-1 exosomes as the vehicle to deliver miRNA- mimetic to NK cells. It was shown that miRNA mimics can be conveyed to NK cells and effectively inhibit target gene expression by exosomes, suggesting its potential for nucleic acid drug delivery.

## 4. Materials and Methods

### 4.1. Ethics Statement

This study was approved by the Ethic Committees of Northwestern Polytechnical University; Ethics approval number: 202102043. The ethics form is shown in the Appendix A.

### 4.2. Cell Lines

The BC cell lines T24 cell, human urothelium immortalized cells SV-HUC-1, K562 cell and HEK 293 cell were obtained from Cell Bank of Chinese Academy of Sciences (BioVector NTCC, Shanghai, China). The three cell lines of T24, SV-HUC-1 and HEK 293 were maintained in DMEM cell culture medium (Gibco, No. 11965-084, New York, NY, USA), supplemented with 100 units/mL penicillin, 0.1 mg/mL streptomycin and 10% (*v/v*) exosome-depleted FBS (SBI, Mountain View, CA, USA), K562 cells were maintained in RPIM-1640 culture medium with the same supplements. All cells were cultured in a humidified 5% CO_2_ atmosphere at 37 °C.

### 4.3. NK Cells Expansion

Human NK cells were expanded from peripheral blood mononuclear cell (PBMC) according to our previous work [50]. Briefly, peripheral venous blood was obtained from healthy donors (10 mL, n = 12) and the PBMCs were separated using lymphocyte separation liquid (Haoyang TBD, Tianjin, China). Genetically modified stimulating cell for NK cell expansion was prepared according to our previous work [50] and cultured in RPIM-1640 cell culture medium containing 100 units/mL penicillin, 0.1 mg/mL streptomycin and 10% (*v/v*) FBS (Gibco, No 10082139, New York, NY, USA). PBMCs were co-cultured with an equal number of irradiation-killed stimulating cells and maintained in RPIM-1640 cell culture medium, supplemented with 100 units/mL penicillin, 0.1 mg/mL streptomycin, 10% (*v/v*) FBS and 100 units/mL IL-2. After 14 days cultivation, the proportion of NK cells in PBMC was evaluated by flow cytometry (BD FACS Calibur, San Jose, CA, USA) using labeled with CD56-PE (BD, Cat No. 561904, New Jersey, CA, USA) and CD3-FITC monoclonal antibodies (mAbs) (BD, Cat No. 561806, New Jersey, CA, USA).

### 4.4. Exosomes Isolation and Identification

The culture supernatant of T24 and SV-HUC-1 cells was collected to isolate exosomes according to the ultracentrifugation method [51]. Briefly, the cell culture supernatant was prepared by initial centrifugation (1 × 10 min, 300× *g*; 1 × 30 min, 2000× *g*) to remove cell debris. The exosomes-containing supernatant was collected and filtered by 0.22 μm filter. The filtrate was ultracentrifuged (1 × 30 min, 100,000× *g*) using a fixed-angle rotor (Ti-70, Beckman Coulter, Inc., Brea, CA, USA) at 4 °C. After washing by PBS, the filtrate was ultracentrifuged (1 × 70 min, 10,000× *g*) at 4 °C. The sediment was stored in −80 °C for the subsequent analysis.

Transmission electron microscopy (TEM) was employed to detect the morphology of exosomes. The exosomes sediment was resuspended in 2% paraformaldehyde aqueous solution and then 3–5 μL exosomes suspension was dripped onto cleaned mica chips. After critical point drying, mica chips were imaged by TEM (FEI, Tecnai G2 Spirit BioTwin, Hillsboro, FL, USA) at 10 kV with a CCD camera (Gatan, Warrendale, PA, USA). Western blot method was used to examine the exosomal markers CD9, CD63 and CD81 on isolated exosomes. Further, the Zetasizer Nano ZS (Malvern Instruments, Malvern, UK) was used to detect the particle size and concentration of isolated exosomes. Each sample was measured three times at room temperature. The protein content of the exosomes was evaluated using BCA assay (BCA Protein Assay Kit, Beyotime, No. P0012-1, Shanghai, China).

### 4.5. Exosomes Uptake Assay

The exosomes uptake assay followed the protocol of our previous work [52]. Briefly, exosomes were labelled by the Exo-Glow Exosome Labeling Kits (SBI, Cat: EXOC300A-1). After resuspending in 500 μL PBS, exosomes (10 μg) were labelled by 50 μL Exo-Green and mixed gently for 10 min. ExoQuick-TC (100 μL) was added into the mixture and incubated for 30 min on ice to stop the labeling reaction. The mixture was ultracentrifuged at 14,000 rpm for 3 min; we then discarded the supernatant and resuspended the exosomes in 100 μL PBS. NK cells (1 × 10^5^) were added into a 12-well plate and co-cultured with Exo-Green labeled exosomes for 24 h. After that, the NK cells were washed by PBS and observed under an inverted fluorescence microscope (ZEISS Inverted Fluorescence Microscope, Axio Observer 3, Oberkochen, Germany) and tested by flow cytometry. The positive rate of the NK cells (only the NK cells that took up labeled exosomes that showed the green fluorescence signal) was calculated.

### 4.6. NK Cell Viability and Cytotoxicity

NK cells (2 × 10^6^ in each group) were treated by T24 exosomes (20 μg/mL) for 24 h to 48 h. After being centrifuged at 1000× *g* for 5 min and washed by PBS, NK cells in each group were resuspended in 1 mL RPMI-1640 media (IL-2 free). NK cell suspension (200 μL) were added into a 96-well plate and repeated for 5 times. Twenty microliters of CCK-8 (Cell Counting Kit-8, Dojindo, Shanghai, China) were added to each well of NK cells and the plate was incubated in a 5% CO_2_ incubator at 37 °C for 2 h. The optical density (OD) value of each well was recorded at 450 nm in a microplate reader (BioTek Synergy-4, Winoosk, VT, USA). The viability of the NK cells was examined.

The cytotoxicity of NK cells was evaluated according to our previous publication [52]. Briefly, NK cells in each group were centrifuged and washed by PBS. Then, NK cells (2 × 10^6^) were re-suspended in RPMI-1640 (IL-2 free) and added into the well of a 96-well plate at 100 μL. Target cell (K562) (4 × 10^5^) which re-suspended in 100 μL medium were added to the same well of NK cell (mix well) to make the effector-to-target ratio (E:T) at 5:1 and co-cultured for 4 h. The wells, containing only NK cell (effector (e) well) and K562 cell (target (t) well), were set and cultured in the same conditions. After 4 h incubation, 20 μL of CCK-8 was added to each well, and the plate was incubated for another 2 h under normal culture conditions. The OD values were recorded at 450 nm. The cytotoxicity was determined by evaluating the rate at which NK cells killed the target cells; the killing rate was calculated using the following equation:Killing rate (%) = [1 − (OD(mix) − ODe)/ODt] × 100%.

### 4.7. NK Cell Apoptosis and Receptor Expression

After T24 exosomes treatment, the apoptosis of NK cell was detected by the Annexin-V/PI double dye method. NK cells in each group were collected and washed by PBS. After being labeled with Annexin V-FITC and PI (AnnexinV-FITC Apoptosis Detection Kit, Cat No. C1062M. Beyotime Institute of Biotechnolog, Shanghai, China), NK cells in each group were analyzed by flow cytometry (BD Calibur, Biosciences, New Jersey, CA, USA).

After T24 exosomes treatment, the NK cells in each group were washed and collected and divided into four groups (1 × 10^6^). Different antibodies of NK cell receptors (NKG2A, NKG2D, NKp30, NKp44, NKp46, and CD226) were used to label each group of NK cells. All the cells were tested by flow cytometry separately and analyzed by Cellquest (BD) software. The information of all antibodies is showed in Table 1.

### 4.8. Exosomal miRNA Profile of T24 Cell

The miRNA profiles of T24 and SV-HUC-1 cell were identified by next-generation sequencing. By comparison to the SV-HUC-1 exosomal miRNA profile, the aberrant expressed miRNAs in T24 cell derived exosomes were identified and verified.

According to the sequencing data, miR-146-3p is the most abundant miRNA in T24 exosomes with a large number of reads and significant up-regulated expression compared to the exosomes of the SV-HUC-1 cell. The other highly expressed miRNAs in T24 exosomes are miR-100-5p, miR-21-5p, miR-21-3p, miR-30a-5p, let-7i-5, miR-221-3p, miR-22-3p, miR-186-5p, miR-28-3p, and miR-378a-3p. The expression of this group of miRNAs in T24 exosomes was verified by the q-PCR method following the instruction of EasyPure miRNA Kit (Transgen Biotech, Beijing, China) and TransScript Green miRNA Two-Step qRT-PCR SuperMix (Transgen Biotech, Beijing, China). The q-PCR circle was implemented in CFX96 Touch qPCR System (Bio-Rad Laboratories, Hercules, CA, USA). Small RNA U6 was used as an internal reference gene, and the 2^−ΔΔCt^ method [53] was used to calculate the relative expression ratio of each miRNA in T24 exosome. The specific primers and the expression of miRNAs in BC exosomes are shown in our previous study [40].

### 4.9. Target Gene Prediction

Four online programs (TargetScan, miRDB, miRTarBase and miRWalk) were employed to predict the potential target genes of these miRNAs in NK cells. The diagrammatic sketch of the interactions of exosomal miRNAs and potential target genes in NK cells was drawn by the Cytoscape software (version 3.6.1) (Appendix A). miR-221-5p and miR-186-5p were selected for further study.

### 4.10. Luciferase Reporter Assay

Using the TargetScan 8.0 database (http://www.targetscan.org/, accessed on 1 January 2020), the binding sites of miRNAs matched to the target genes were predicted. Figure 5 depicts the algorithm-predicted binding site of miRNA to the target genes. Luciferase reporter assay was employed to confirm the combination of miRNA and target genes. Following the manufacturer’s protocol, the position of the sequence region, containing the putative binding sequence of each miRNA, was inserted into a luciferase reporter vector pGLO-basic (Promega, Madison, WI, USA). The mutated sequence of the target gene was constructed into a pGLO-basic vector. The sequences constructed in the reporter vector and mutant vector were confirmed by sequence analysis. pRL-TK Renilla Luciferase Reporter vector (Promega, Madison, WI, USA) was used as an internal control vector. HEK 293 cells were seeded into 96-well plates with the confluent rate about 80% and co-transfected with either reported vector (0.01 mg/well each), miRNA mimic/mutated vector (10 nM) and internal control vector. After 48 h co-transfection, luciferase activities were measured by a microplate reader (Infinite M1000, TECAN, Männedorf, Swiss). Renin luciferase activity was defined as the standardization of firefly luciferase activity.

### 4.11. NK Cell Transfection via Overexpression Target miRNAs in SV-HUC-1 Exosome

Due to the very low efficiency of transfecting NK cell by conventional transfection methods, we attempted to use exosomes as the vector to carry miRNA and transfect NK cells. The mimics of miR-186-5p (Cat no. miR10000456-1-5; RiboBio, Guangzhou, China), miR-221-5p (Cat no. miRB0004568-2-1; RiboBio, Guangzhou, China) and the NC mimic has-miR-NC (Cat no. miR1190315051351, RiboBio, Guangzhou, China) were purchased from Guangzhou RiboBio Co.,LTD. The miRNAs were loaded into exosomes via electroporation according to the protocol by Lamichhane et al. [54]. Briefly, SV-HUC-1 derived exosomes (5 µg) were mixed with each kind of miRNA (5 µg) in 50 µL electroporation buffer (1.15 mM K_3_PO_4_, pH: 7.2, 25 mM KCl, 21% OptiPrep). Electroporation was carried out using Gene Pulser/Micropulser Cuvettes (Bio-Rad, Cat no.165-2089, Hercules, CA, USA) in a GenePulser Xcell electroporator (Bio-Rad, Hercules, CA, USA). Electroporation was carried out at 400 V and 125 µF with three pulses. Then the samples were transferred into a tube (0.5 mL) and 1 mM EDTA was added. After incubation at RT for 15 min, the samples were centrifuged at 100,000× *g* at 4 °C for 5 min to remove buffer and unincorporated miRNA. Then, following 4 h treatment, the content of target miRNAs mimic and NC mimic in transfected NK cells was verified by the q-PCR method. The specific primers of miR-186-5p and miR-221-5p are shown in Table 2.

### 4.12. The Verification of Target Genes in NK Cell

After miRNAs carrying exosomes treatment for 48 h, the expression of DAP10, CD96, Foxo1, perforin and NKG2D in NK cells was analyzed in the protein level. The details of antibodies are shown in Table 1.

Using the Western blot method, the protein expression was evaluated. After being washed with PBS, NK cells in each group (2 × 10^7^ each) were lysed by RIPA lysis buffer (Cat No. P0013C, Beyotime Institute of Biotechnology) supplemented with protease inhibitor (Cat No. P1051, Beyotime Institute of Biotechnology) on ice for 10 min. The lysate of the cells was quantified by BCA assay (BCA Protein Assay Kit, Cat No. P0012S, Beyotime Institute of Biotechnology). Briefly, the lysate was separated by 12% Tris-glycine gels in equal amounts, and then transferred onto polyvinylidene fluoride (PVFD) membranes. Under gentle shaking, the blots were first blocked in 5% nonfat milk for 1 h. After washing in TBST three times, the blots were incubated with primary antibodies overnight at 4 °C, and β-actin was employed as internal control. After washing in TBST three times, the blots were incubated with the secondary antibodies at room temperature for 1 h with gentle shaking. The blots were washed again with TBS, and then immersed in the luminous liquid (EasySeeVR Western Blot Kit, Cat No. DW101-02, TransGen Biotech, Beijing, China) for 1 min. The signals were detected by exposure of the film in a chemiluminescence device (Tanon-5200Multi, Shanghai, China).

### 4.13. Data Analysis

Statistical analyses were performed by using Graphpad Prism 8 software. The data were presented as the mean ± SD. The results were analyzed by using analysis of Student’s *t* test and Analysis of Variance (ANOVA). Multiple comparisons were performed using the LSD test to evaluate significant differences between the groups. Statistical significance was defined as *p* < 0.05.

## 5. Conclusions

This work indicated that T24 cell-derived exosomes can induce NK cell dysfunction by impairing viability, and inhibit the cytotoxicity of NK cells. The important functional receptors NKG2D, NKp30, and CD226 on NK cells and the secretion of perforin and granzyme-B were downregulated following treatment with T24 cell-derived exosomes. miR-221-5p and miR-186-5p are the critical exosomal miRNAs in T24 cell-derived exosomes and can effect NK cell function by interfering with the stability of mRNAs of the DAP10, CD96, and perforin genes. Additionally, exosomes can be employed as the nucleic acid carrier to delivery miRNA into NK cells efficiently.

## Figures and Tables

**Figure 1 ijms-23-15177-f001:**
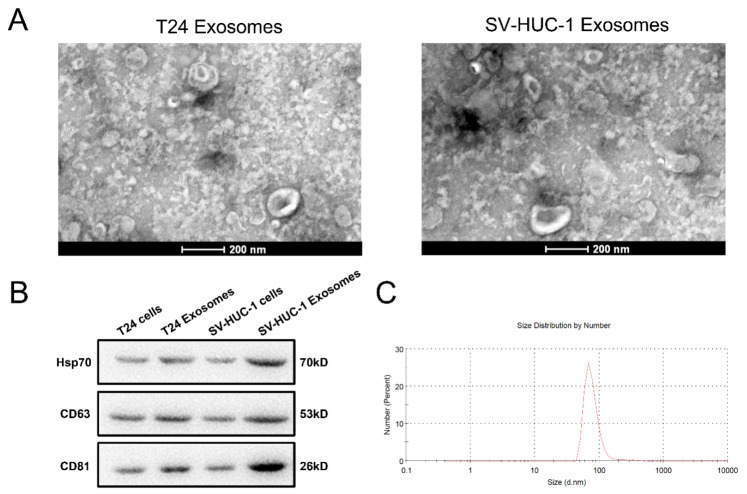
Identification of exosomes. (**A**): Exosome morphology under TEM. Exosomes were re-suspended and fixed using 2% *w/v* paraformaldehyde aqueous and added onto cleaned mica chips, critical point drying and imaged by TEM. (**B**): WB analysis of the exosome markers (Hsp70, CD63, CD81). (**C**), Particle size distribution of exosome; exosomes were diluted by 1 × PBS, and measured by a Zetasizer Nano (n = 3).

**Figure 2 ijms-23-15177-f002:**
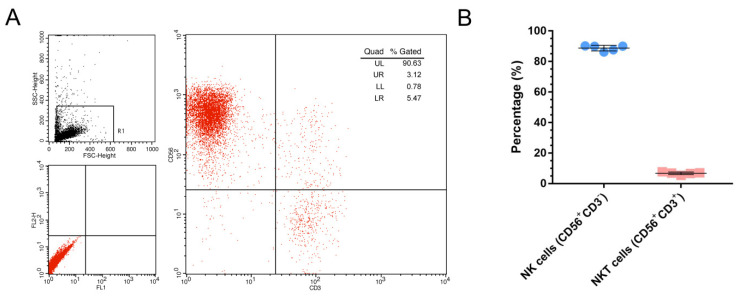
The proportion of NK cells. (**A**): The representative flow cytometry data of NK cells (CD56^+^CD3^−^) proportion. (**B**): The statistical results of NK cells (CD56^+^CD3^−^) and NKT (CD56^+^CD3^+^) cells proportion (n = 5).

**Figure 3 ijms-23-15177-f003:**
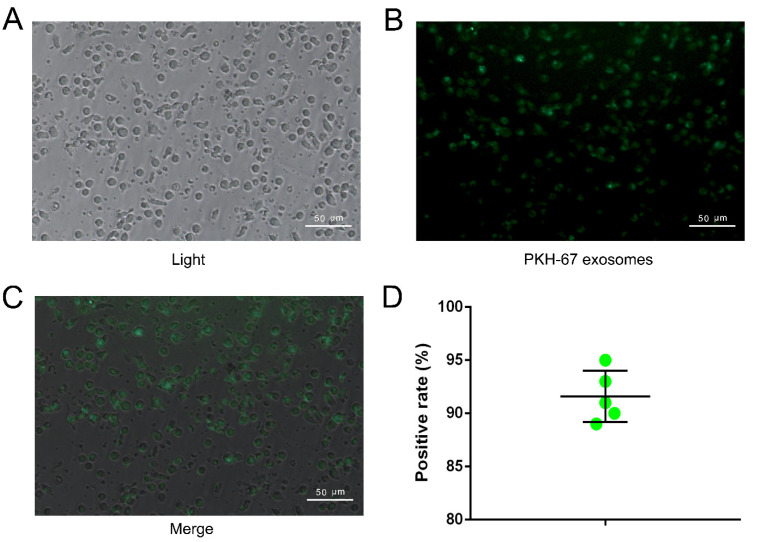
The exosomes uptake efficiency of NK cells. (**A**): The NK cells in light field. (**B**): The NK cells in fluorescence field, through uptake the Exo-Glow labeled T24 exosome, NK cell showed green fluorescent. (**C**): The positive rate of NK cells which showed green fluorescent tested by flow cytometry (n = 5). (**D**): The positive rate of NK cells.

**Figure 4 ijms-23-15177-f004:**
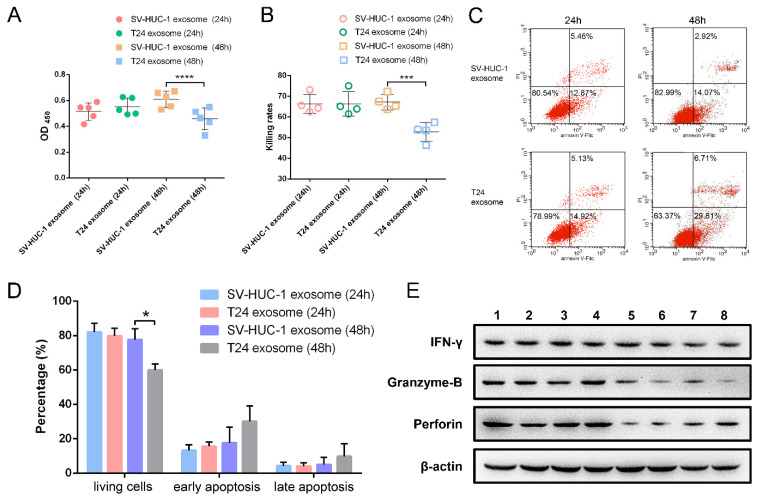
The effects of T24 exosome on NK cells viability, cytotoxicity, apoptosis, and cytokines expression. (**A**): The viability of NK cells after 24 h and 48 h T24 exosome treatment (n = 5). **** *t* test, *p* < 0.0001; *** *t* test, *p* < 0.001. (**B**): The killing rate of NK cells after 24 h and 48 h T24 exosome treatment (n = 4). (**C**): The representative flow cytometry data of NK cell apoptosis. (**D**): The bar graph showed the percentage of viable (Annexin V^−^/PI^−^), early apoptotic (Annexin V^+^/PI^−^), and late apoptotic/necrotic (Annexin V^+^/PI^+^) cells in each group (n = 5). T24 exosome treatment significant decrease the proportion of living NK cells after 48h treatment. * *t* test, *p* < 0.05. (**E**). Western blot analysis of cytokines (IFN-γ, perforin, granzyme-B) secretion of NK cells. Line 1–4, treated by SV-HUC exosomes, line 5–8, treated by T24 exosomes. The expression of granzyme-B and perforin were decreased after T24 exosome treatment.

**Figure 5 ijms-23-15177-f005:**
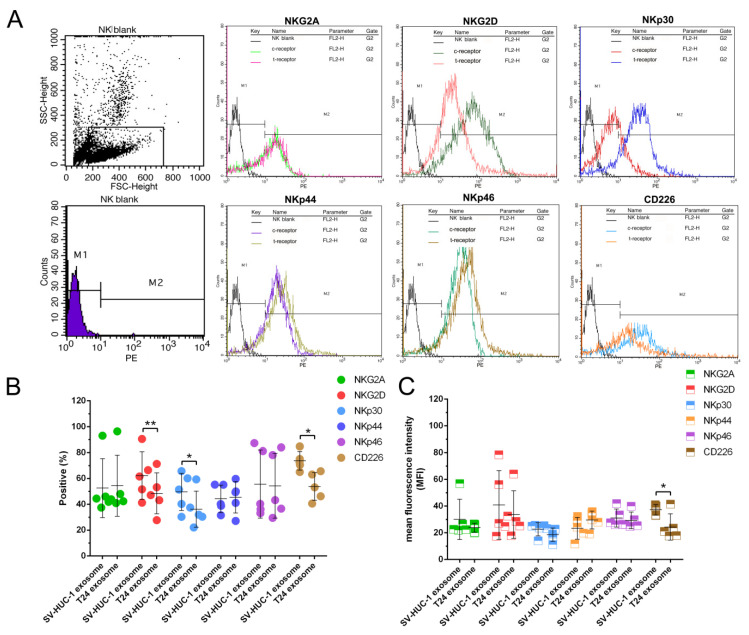
The effects of T24 exosome on the expression of NK cells functional receptors. (**A**): The representative flow cytometry data of the expression of NK cells functional receptors (NKG2A, NKG2D, NKp30, NKp44, NKp46 and CD226). (**B**): The bar graph showed the positive rate of NK cells functional receptors (n = 5); (**C**): The bar graph showed the mean fluorescence intensity (MFI) of NK cells functional receptors (n = 5); The positive rate of NKG2D, NKp30 and CD226 were reduced after T24 exosome treatment; The MFI of CD226 was reduced after T24 exosome treatment as well. * *t* test, *p* < 0.05; ** *t* test, *p* < 0.01.

**Figure 6 ijms-23-15177-f006:**
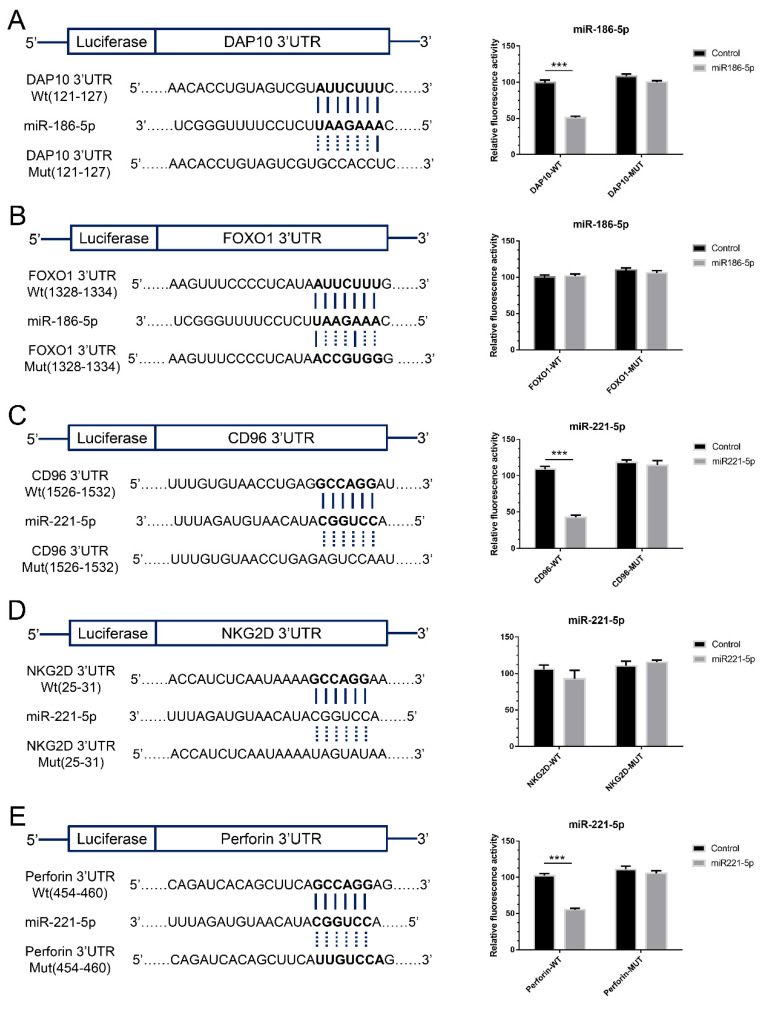
The pairing relationship of miRNAs and target genes. (**A**): The binding site and Dual-Luciferase reporter assay of miR-186-5p matched to DAP10 genes. (**B**): The binding site and Dual-Luciferase reporter assay of miR-186-5p matched to FOXO1 genes. (**C**): The binding site and Dual-Luciferase reporter assay of miR-221-5p matched to CD96 genes. (**D**): The binding site and Dual-Luciferase reporter assay of miR-221-5p matched to NKG2D genes. (**E**): The binding site and Dual-Luciferase reporter assay of miR-221-5p matched to Perforin genes. *** *t* test, *p* < 0.001.

**Figure 7 ijms-23-15177-f007:**
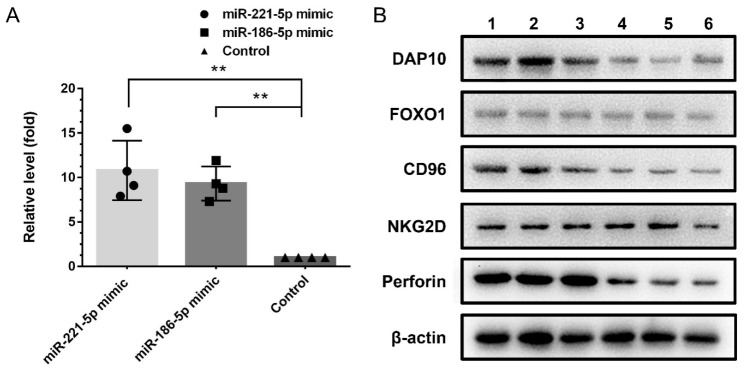
The transfection and interfere efficiency of miRNA mimics in NK cells. (**A**): qPCR method to verify the transfection efficiency. It indicated the miR-186-5p and miR-221-5p have been effective transfected into NK cells (**B**): Western blot analysis of the expression on the target genes (DAP10, Foxo1, CD96, NKG2D, perforin) in NK cells after mimic transfection; Line 1–3, transfected by NC mimic using SV-HUC exosomes, line 4–6, transfected by mimic of miR-186-5p and miR-221-5p using SV-HUC exosomes. The expression of DAP10, CD96 and perforin were down-regulated after mimic transfection. ** *t* test, *p* < 0.01.

**Table 1 ijms-23-15177-t001:** The information of all antibodies.

Antibody	Brand	Catalog Number.
PE Mouse Anti-human NKG2A	R&D	FAB1059P
PE Mouse Anti-human NKG2D	BD Pharmingen	554680
PE Mouse Anti- human NKp30	BD Pharmingen	558407
PE Mouse Anti-human NKp44	BD Pharmingen	558563
PE Mouse Anti-human NKp46	BD Pharmingen	557991
PE Anti-human CD226 [DX11]	abcam	ab33337
Hsp70	abcam	ab2787
CD63	abcam	ab134045
CD81	abcam	ab79559
IFN-γ	abcam	ab267369
Granzyme-B	abcam	ab255598
Perforin	abcam	ab256453
DAP10	Santacruze	sc-374196
FOXO1	abcam	ab179450
CD96	abcam	ab264416
NKG2D	abcam	ab96606
β-actin	abcam	ab8226

**Table 2 ijms-23-15177-t002:** Primers sequences used for quantitative real time PCR of miRNAs.

miRNA ID	Accession Number	Forward Primer Sequence (5′–3′)	Tm	%GC
hsa-miR-221-5p	MIMAT0004568	CGCGACCTGGCATACAATGT	61.6	54.55%
hsa-miR-186-5p	MIMAT0000456	ATGCGCGCCAAAGAATTCTCC	62.78	52.38%

## Data Availability

The data that support the findings of our study are available from the corresponding author upon reasonable request.

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
