# Peer review of "miR-221-5p and miR-186-5p Are the Critical Bladder Cancer Derived Exosomal miRNAs in Natural Killer Cell Dysfunction"

_ijms, 2022, doi:10.3390/ijms232315177_

Round 1

Reviewer 1 Report

The work describes how exosomes derived from BC cells can influence the NK cells behavior resulting in the immuno-escape. 

I believe that the manuscript sounds well and could involve a wide range of researchers. However, I have some suggestions/improvements:

1)In the Introduction section I believe that two sentences lack references (lines 55 and 59....i.e.PMID: 34496230 for the first and  PMID: 35141731 for the second line)   

2)The flow cytometry representation in Figure 5A is not optically good. It could be better if the histograms would be smooth without the "zig-zag" of the Gaussian line. 

3)The authors analyzed possible differences in IFN-gamma relapse after treatment on NK with T24 exosomes. Despite no differences being found, I believe that this should be confirmed also in the second experiment, that is in the treatment of NK cells with modified exosomes charged with miRNAs. I believe that this could enforce the first data. 

Reviewer 2 Report

The submitted manuscript is a very interesting example of the article taking into consideration the NK cell activity altered by exosomal miRNAs. The introduction part including a description of bladder cancer and exosomes is well prepared. The Reviewer is really amazed by the Authors' scientific rigor in conducting and reporting their research. 

Material and methods are well and properly described. Statistical tests were chosen correctly.

The Results are properly described.

The Discussion section is prepared in a thoughtful way and a sufficient number of articles was cited.

The Reviewer found in some paragraphs doubled spaces.

Summarizing, I recommend this manuscript for publication when the Authors correct the abovementioned issues in the article.

Reviewer 3 Report

Ting Huyan and collaborators investigated the role of T24 derived exosome on NK cell dysfunction. They observe that the viability and the cytotoxicity of NK cell were inhibited by T24 derived exosome. This was correlated with downregulated NKG2D, NKp30, and CD226 expression and impaired secretion of perforin and granzyme-B. In addition, the authors found that miR-221-5p and miR-186- 5p were the critical exosomal miRNAs in T24-EXO, effecting NK cells function by interfering the stability of mRNAs of DAP10, CD96, and perforin gene.

Overall, this manuscript is interesting. However, the following questions should be addressed:

     1)     All the data are based on BC cell line T24. Did the authors confirm the data in others BC cell lines? This should be done using 5637, RT4 or others BC cells line.

     2)     Did the authors investigate the role of miR-221-5p and miR-186- 5p in Exosomes from luminal-like and basal-like BC? Is there any difference?

     3)     Is there any evidence of the role miR-221-5p and miR-186- 5p in specimens from BC patients?

     4)     Fig 7B: Western blot analysis showed that the expression NKG2D in NK cells after mimic transfection is not affected. In contrast, in Fig 5A, the data of flow cytometry indicated that the NKG2D expression was significant effected by T24-EXO. It looks like a sort of contradiction. How to explain these different results?

Round 2

Reviewer 3 Report

All data were obtained using just one BC cell line, T24. The conclusions should be supported by results obtained in at least another different BC cell line. At this stage, the manuscript is of interest, but the results are too preliminary. Most of the raised questions remain to be addressed.

Author Response

Point 1: All data were obtained using just one BC cell line, T24. The conclusions should be supported by results obtained in at least another different BC cell line. At this stage, the manuscript is of interest, but the results are too preliminary. Most of the raised questions remain to be addressed.

Response 1: We are highly grateful for your careful critique and valuable comment. Firstly we strongly agree with your suggestion and the current research data is obtained based on T24 cells. We would love to be able to conduct a more in-depth study based on your suggestion. However, as we explained previously, we are not yet able to obtain more data on bladder cancer cell lines in a short period of time. Furthermore, although the miRs were derived from a comparison of the T24 exosomes and SV-HUC-1 exosomes miRNA groups, the article focused on the effects of the miRNAs themselves on NK cells and their cellular origin did not affect their effects on NK cells. Finally, we have revised the following areas to make our article more rigorous according to your suggestions:

(1) the title of this manuscript was changed to “miR-221-5p and miR-186-5p are the critical bladder cancer exosomal miRNAs in natural killer cells dysfunction”;

(2) The “T24 cell-derived exosomes” was used to replace the BC exosome to make the definition more rigorous in the reviesd manuscript.

We do hope our explanation and revision will bring some light to this question and thank you again for your comment.

.

Round 3

Reviewer 3 Report

The new version of this manuscript is sufficiently improved.

Author Response

Point 1: The new version of this manuscript is sufficiently improved.

Response 1: Thank you very much for your help in the process of reviewing and revising the paper, and we will definitely conduct an in-depth study based on your suggestions in our future research. Thank you again for your hard work.
